# Discharge pharmacotherapy for Type 2 diabetic inpatients at two hospitals of different tiers in Zhejiang Province, China

**Xiaofang Yu** [1]*, **Long Zhang** [1©], **Rongbin Yu** [2©], **Jiao Yang** [3©], **Saifei Zhang** [1]

**1** Department of Endocrinology in Ningbo Medical Treatment Centre Lihuili Hospital, Ningbo, Zhejiang Province, China, **2** Department of Preventive Care and Medical Insurance in Ningbo Medical Treatment Centre Lihuili Hospital, Ningbo, Zhejiang Province, China, **3** Medical Department in Simen Hospital, Yuyao City, Ningbo, Zhejiang Province, China

© These authors contributed equally to this work.

* Yuxf210@126.com

**Data Availability Statement:** All relevant data are within the manuscript.

**Funding:** No funding or financial support from any source was received for this analysis.

## Abstract

### Objects

To look into the discharge pharmacotherapy for type 2 diabetics admitted to two general hospitals of different ranks and inspect current real-world management of discharge pharmacology and its related factors.

### Methods

Type 2 diabetics admitted to a tertiary general hospital (Ningbo Medical Treatment Centre Lihuili Hospital, LHLH) or a secondary general hospital (Simen Hospital, SMH) for intensification of their anti-diabetics were included for retrospective analysis. Patients' demographics, clinical characteristics, admission diabetes therapy and discharge diabetes pharmacology were analyzed and compared among patients in each hospital as well as between two hospitals.

### Results

391 patients from LHLH and 164 patients from SMH were included for analyzing. Compared with patients from LHLH, patients from SMH were older, more illiterate and had higher HbA1c concentrations. While there was a nearly equal split of oral anti-diabetes drugs (OADs)-only and Insulin treatment in LHLH's discharge pharmacotherapy, insulin treatment dominated SMH's. Basal-and-bolus insulin assumed the majority of SMH's insulin regimens but only accounted for less than 20% of LHLH's. The principal discrepancy in OADs-only treatment existed in the utilization of newer classes of OADs. Cost and body mass index (BMI) were the main differentiating factors among OADs-only treatments while duration, BMI and HbA1c differ among insulin treatments at LHLH. Clinical characteristics didn't significantly differ among OADs-only treatments and HbA1c was the only differentiating factor among insulin treatments at SMH. Overall, hospital, duration, HbA1c, and vascular diseases were main factors that affect discharge pharmacology.

**Competing interests:** The authors have declared that no competing interests exist.

## Conclusions

Great disparities exist in the discharge pharmacotherapy at two hospitals. Diabetes management is mostly glucose-oriented at SMH while multifactorial considerations were reflected in LHLH's discharge pharmacotherapy. Besides differences in patients' demographics, medication availability and diagnosis of early-stage vascular complications, lack of practical algorithm for discharge management in T2DM may be the underlying deficiency and a key part for future improvement.

## Introduction

Type 2 diabetes mellitus (T2DM) constitutes the majority of adult diabetes and has become a growing contributing part to adolescent and childhood diabetes [1–4]. Pathophysiologic abnormalities in T2DM vary greatly among different clinical phenotypes as well as along different stages of the disease [5–7]. It's ideal to make anti-diabetes pharmacotherapy pathophysiologically personalized in T2DM but difficult to realize [8]. Besides, multifactorial considerations are required of T2DM pharmacotherapy like hypoglycemic potency, safety, cost, patients' preference and so on [9]. As a result there is no 'best' treatment for T2DM [8, 10]. Guidelines of standardized management for T2DM developed by diabetes societies around the world provide a practical way of stepwise arrangement of anti-diabetic medications [2, 9, 11, 12]. So far the Chinese Diabetes Society has published five versions of guidelines with the latest one released in 2018 [2] and has achieved big progression in promotion of standardized treatment. Yet one aspect the Chinese guideline neglects is the discharge management for T2DM patients [2], which is a critical part of successful hospitalization for T2DM [11, 13]. Unlike the generally accepted in-hospital management of hyperglycemia [14, 15], there is little consensus on algorithms for discharge transition [16, 17]. It's conceivable that there would be greater discrepancies in discharge pharmacotherapy for T2DM.

Another obstacle to standardized diabetes management comes from imbalanced distribution of medical resources (hospital facilities, faculty competence, medication availability, etc.), which is mostly representative in China. One major challenge in Chinese diabetes management derives from it diverse demographics and imbalanced distribution of medical service [18]. Even within the same administrative region the urban areas and the rural areas could show great differences in disease awareness, treatment and control [19]. The promotion of hierarchical diabetes treatment is one measure taken by Chinese government to improve this situation [18]. Accordingly, regional longitudinal health alliances (RLHAs) are constructed. A RLHA usually consists of a leading tertiary general hospital, several secondary hospitals and many community healthcare centers (According to the national accreditation system for hospitals in China, there are three levels of providers with each level further classified as Class A/B/C. Tertiary hospitals accredited as Class 3A are top providers, with more beds and more proficient staff as well as most advanced administration and medical technology, followed by Class 3B/3C, then Class 2A/2B/2C and so on till Class 1C as the primary settings only with basic facilities). RLHAs are basic functional units of hierarchical diabetes management. The construction of RLHAs is a multi-sectoral collaboration involving administrative, financial, educational as well as clinical affairs. Multiple measures are taken in this process including promotion of dual-referral systems, organized continuing-education classes and short-term training programs provided by higher ranks of institutions, supplement to medication lists at

inferior institutions, periodical and on-demand consultations and so on. In the long run, better implementation of RLHAs will realize: 1. better dissemination of standardized management of uncomplicated T2DM and improved disease management in primary settings; 2. creation of more effective health care delivery systems and realization of orderly dual-referral for complicated diabetes.

Ningbo City, Zhejiang Province is of the first batch of cities that carried out the policy three years ago. Ningbo is a port city with 800 million inhabitants in southeast China. Ningbo Medical Treatment Centre Lihuili Hospital (LHLH) is one of the four tertiary general hospitals in Ningbo, accredited as Class3A level provider. By the policy of regional medical alliance, LHLH accepts referrals from 5 secondary hospitals but practically serves the whole population in Ningbo. Among the 5 secondary institutions, only Simen Hospital (SMH) is not located in the urban areas, but in a town named Yuyao which is about 60 kilometers from Ningbo City. In the constitution of the LHLH-leading health alliance, SMH is a secondary unit with another 4 community-based health services subordinate to it. Among the 5 secondary institutions, SMH is special in two meanings. For one, the population it serves differs from all the other 4 institutions and gives a good chance to exemplify management of diabetes in the rural population; for another, it assumes a more typical connecting role as a secondary institution in the health alliance since currently urban residents have a preference to skip the orderly hierarchical system and directly seek medical service at tertiary hospitals like LHLH, possibly rendering another four hospitals less responsibilities (another article discussed in detail about the situation of the dual-referral system in Ningbo, not published yet).

Taken the contexts under consideration, we targeted a subgroup of T2DM hospitalized at LHLH or SMH for intensification of their diabetes therapies as representative patients to investigate the real-world management of uncomplicated T2DM, with a focus on the discharge anti-diabetic pharmacology. The primary object of this study is to inspect the discharge pharmacotherapy in T2DM at hospitals of two hierarchical levels and to detect possible discrepancies and the underlying influencing factors. The secondary object is to get a sketch of the real-world pharmacological management of T2DM and to get a preliminary understanding of the question 'where are we along the road of promoting standardized diabetes management' [2, 9].

## Materials and methods

The analysis was performed on inpatients' medical records. Patients' records were accessed respectively from the hospital information system (HIS) at LHLH and SMH during May and August 2019 by trained data collectors and then analyzed by another analyst. No identifiable data of patients were accessible to all researchers and no individual interactions with patients were made. The study is approved by ethics committees from the hospitals (the Ethics Committee of Ningbo Medical Treatment Centre Lihuili Hospital and the Ethics Committee of Fourth Hospital, Yuyao/Simen Hospital) and consent forms are waivered for this retrospective and anonymized analysis.

All patients that were admitted to LHLH from Jan 1, 2018 to Apr 30, 2019 or to SMH from Jan 1, 2017 to Apr 30, 2019 and met the inclusion criteria without following conditions listed as exclusion criteria were eligible for further analysis. Inclusion criteria are: 1. the principal discharge diagnosis being T2DM or its chronic complications, or the main discharge diagnosis ICD-10 coding being E11. 2. age $\geq$ 18 years. 3. newly diagnosed diabetes: patients diagnosed as T2DM for the first time, no matter whether they provide a long history of osmotic symptoms or not. Patients previously knew themselves as 'diabetes' or 'hyperglycemic' were not included as 'newly diagnosed' in this study even with a duration as short as less than one year, be them treated or not. Patients with following conditions were excluded: 1. Diabetic

emergencies like diabetic ketoacidosis or hyperglycemic hyperosmolar status. 2. Clinically relevant hepatic disease (with plasma transaminase $\geq$ 2.5 times the upper limit, whatever the causes are; or, cirrhosis, with or without biochemical abnormalities). 3. Kidney insufficiency (with serum creatinine $\geq$ 1.5 mg/dl or eGFR<45mL/min.1.73m$^2$) [2]. 4. Infections with systemic reactions. 5. Patients who were expecting any surgical operations in the near future. 6. Corticoid treatment. 7. Pregnancy or lactation.

Inpatients' medical records were scrutinized. The following information of eligible patients was exported to Microsoft Excel for further analysis including: 1.Basic information such as hospitalization number, medical insurance state, hospital length of stay (LOS), admission and discharge diagnoses. 2. Demographics: gender, age, literacy, body weight, BMI (only available at LHLH). 3. Disease characteristics: duration (of T2DM), admission HbA1c concentration, (history of) hypertension, macro-vascular diseases including: (history of) coronary heart disease (mostly based on history at both hospitals, except for some patients at LHLH were newly diagnosed as coronary heart disease via computed tomography angioplasty, but no patients went through percutaneous coronary intervention), history of cerebral vascular disease (mostly based on history in medical record and without differentiation between ischemic and hemorrhagic ones; except for a few patients with suspected symptoms got diagnosed as ischemic disease through brain magnetic resonance imaging, but no patients needed thrombolytic therapy or intravascular interventions), and peripheral artery diseases (represented by atherosclerosis in lower limbs arteries and/or carotid arteries, routinely screened with Doppler ultrasonography at LHLH). Micro-vascular diseases represented by diabetic kidney diseases (DKD, diagnosed as urinary albumin creatinine ratio, UACR>30 mg/g and/or estimated glomerular filtration rate, eGFR <60ml.min$^{-1}$.[1.73m$^2$]$^{-1}$ [2]; UACR only doable at LHLH.) and diabetic retinopathy (DR, diagnosed according to international clinical grading standard for diabetic retinopathy [20] at LHLH but mostly based on history at SMH). 4. Admission diabetes therapy (None, OADs only, Insulin only, insulin +OADs), and discharge pharmacotherapy: OADs-only treatment (For convenience we included glucagon-like peptide-1 receptor agonist, GLP-1RA, the injectable non-insulin therapy below the class of 'OADs' in the following analysis.) or, insulin treatment (premix insulin, basal insulin, basal-and-bolus insulin). Combinations of OADs in the OADs-only treatment and doses of insulin in the insulin treatment were specifically recorded.

The data were generally described as mean±standard deviation (SD) for continuous variables and count (absolute and percentage) for categorical variables. Student's test was used to analyze between groups of continuous variables. Multiple groups of continuous data were analyzed with One-way Anova tests and the comparison among groups were performed with S-N-K test. Categorical variables were analyzed with $\chi^2$ test with $p$<0.05 considered statistically significant. Partitions of $\chi^2$ method was used to compare among groups of categorical data ($\alpha^{'}$ = [2*$\alpha$]/ [k* [k-1]+1]). Generalized linear models (GLM) and generalized linear mixed models (GLMM) were applied to study influences of discharge pharmacology. GLMM was run with glmer() function from Package lme4 in R statistics. Package TableOne was used to calculate the odds ratio (OR) and 95% confidential interval (CI). The soft packages used in this analysis were IBM SPSS (version 23.0) and R (version 3.6.1).

## Results

### Demographical and clinical characteristics of patients from LHLH and SMH

Totally 391 patients at LHLH and 164 patients at SMH met the inclusion criteria. Table 1 gives the demographical features and clinical characteristics of the patients. Compared with patients from LHLH, patients from SMH were older, female-preponderant in prevalence, less literate

**Table 1. Demographical features and clinical characteristics of the patients at LHLH and SMH.**

| | LHLH | SMH | $t/\chi^2$ | *P* value |
|---|---|---|---|---|
| N | 391 | 164 | | |
| Age (years) | 58.4±12.23 | 61.3±10.65 | -2.79 | 0.006 |
| Male (n, %) | 231 (59.1) | 68 (41.5) | 14.428 | <0.01 |
| Illiterate (n, %) | 52 (13.3) | 88 (53.7) | 99.776 | <0.01 |
| Insurance-covered (n, %) | 225 (57.5) | 74 (45.1) | 7.175 | <0.01 |
| Newly diagnosed diabetes † (n, %) | 74 (18.9) | 54 (32.9) | 12.764 | <0.01 |
| Duration of diabetes (years) | 9.1±7.28 | 9.2±8.07 | -0.125 | 0.901 |
| Hospital LOS (days) | 8.2±3.82 | 8.3±2.95 | -0.366 | 0.715 |
| Admission HbA1c (%) | 9.51±2.68 | 10.42±2.05 | -4.324 | <0.01 |
| BMI (kg/m$^2$) | 23.8±3.70 | / | / | / |
| Admission diabetes therapy | | | | <0.01 |
| None (n, %) | 96 (24.6) | 84 (51.2) | | |
| OADs (n, %) | 220 (56.3) | 69 (42.1) | | |
| Insulin only (n, %) | 39 (10.0) | 5 (3.0) | | |
| OADs and insulin (n, %) | 36 (9.2) | 6 (3.7) | | |
| Macro-vascular diseases | | | | |
| Carotid artery atherosclerosis (n, %) | 203 (51.9) | 24 (14.6) | 66.445 | <0.01 |
| Lower limb atherosclerosis (n, %) | 194 (49.6) | 12 (7.3) | 88.570 | <0.01 |
| Coronary heart diseases (n, %) | 37 (9.5) | 7 (4.3) | 4.271 | 0.039 |
| Cerebral vascular diseases (history) (n, %) | 27 (6.9) | 11 (6.7) | 0.007 | 0.933 |
| Micro-vascular diseases | | | | |
| Diabetic kidney disease (n, %) | 47 (12.0) | 12 (7.3) | 2.690 | 0.101 |
| Diabetic retinopathy (n, %) | 28 (7.2) | 1 (0.6) | 10.014 | <0.01 |
| Hypertension (n, %) | 207 (52.9) | 102 (62.2) | 4.009 | 0.049 |

† newly diagnosed diabetes: patients diagnosed as diabetes for the first time, no matter whether they provide a long history of osmotic symptoms or not. Patients previously knew themselves as 'diabetes' or 'hyperglycemic' were not included as 'newly diagnosed' even with a duration as short as less than one year, be them treated or not.

and had poorer insurance-coverage. The mean duration of diabetes didn't differ between two hospitals. Admission HbA1c concentration was significantly higher at SMH. Nearly 1/3 of the patients at SMH, and 1/5 at LHLH didn't know they were diabetic before. About 20% and 5% of the known T2DM at SMH and LHLH, respectively, were untreated. Among those with previous diabetes treatment, most were receiving 'OADs only' treatment at SMH; more patients had initiated insulin treatment at LHLH.

Half of the patients at LHLH had peripheral artery atherosclerosis but the diagnostic rate was much lower at SMH. Coronary heart disease was twice more common in patients at LHLH. The prevalence of cerebral vascular diseases was similar at two hospitals. Hypertension was much more prevalent in patients from SMH than patients from LHLH. More patients were diagnosed with DKD at LHLH, but it didn't reach statistical significance. Diabetic retinopathy was scarcely diagnosed at SMH.

## Discharge pharmacology for type 2 diabetes at LHLH and SMH

Table 2 displays the discharge pharmacotherapy at the two hospitals. OADs-only and insulin treatment assumed a nearly equal share at LHLH and only 4 patients (1.1%) were treated with diet-only. Insulin treatment assumed three-quarters of SMH's discharge pharmacotherapy and none was treated with diet-only. Traditional OADs were similarly distributed at two

**Table 2. Treatment regimens at discharge in patients from LHLH and SMH.**

| | | LHLH (n, %) | SMH (n, %) | $t/X^2$ | *P* value |
|---|---|---|---|---|---|
| **OADs** | Metformin | 284 (72.6) | 114 (69.5) | 0.56 | 0.46 |
| | Sulfonylureas | 92 (23.5) | 40 (24.4) | 0.05 | 0.83 |
| | AGI | 226 (57.8) | 69 (42.1) | 11.48 | <0.01 |
| | TZD | 3 (0.8) | 4 (2.4) | 2.59 | 0.11 |
| | Incretin therapy | 136 (34.8) | 4 (2.4) | 39.40 | <0.01 |
| | DPP-4i | 86 (22.0) | 4 (2.4) | 32.52 | <0.01 |
| | *GLP-1 RA | 50 (12.8) | 0 (0.0) | 23.10 | <0.01 |
| | Glinide | 28 (7.1) | 18 (11.0) | 2.12 | 0.137 |
| | SGLT-2i | 16 (4.1) | 0 (0.0) | 6.91 | <0.01 |
| **Combinations of OADs** | TOTAL | 171 (43.7) | 43 (26.2) | 3.24 | 0.34 |
| | Monotherapy | 29 (17.0) | 4 (9.3) | 1.42 | 0.23 |
| | Dual therapy | 81 (47.4) | 20 (46.5) | 0.001 | 0.98 |
| | Triple therapy | 51 (29.8) | 18 (41.9) | 2.62 | 0.11 |
| | Quadruple therapy | 10 (5.8) | 1 (2.3) | 0.83 | 0.36 |
| **Insulin regimens** | Total (n, %) | 216 (55.2) | 121 (73.8) | 70.5 | <0.01 |
| | Basal insulin | 93 (43.1) | 38 (31.4) | | |
| | Daily dose (u/kg) | 0.21±0.09 | 0.20±0.06 | 0.40 | 0.69 |
| | Premix insulin | 84 (38.9) | 10 (8.3) | | |
| | Daily dose (u/kg) | 0.49±0.18 | 0.46±0.13 | 0.57 | 0.57 |
| | Bolus and basal insulin | 39 (18) | 73 (60.3) | | |
| | Daily dose (u/kg) | 0.64±0.14 | 0.66±0.11 | 0.72 | 0.47 |
| | Bolus/basal ratio | 1.46±0.60 | 1.44±0.57 | 0.28 | 0.78 |

*injection formula; AGI, α glycosidase inhibitor; TZD, thiazolidinedione; GLP-1 RA, glucagon-like peptide-1 receptor agonist; DPP-4i, dipeptidyl peptidase-4 inhibitor; SGLT-2i, sodium-glucose co-transporter-2 inhibitor.

hospitals with the exception of α glycosidase inhibitor (AGI) which was more common at LHLH. New classes of OADs were still rare in SMH's treatment. Of the patients receiving OADs-only, a majority of them were on dual or triple therapy at SMH. Discharge pharmacotherapy diverged in distribution of insulin regimens. Most of patients remained on basal-and-bolus insulin at SMH while Basal and premix insulin constitute the majority of LHLH's insulin treatment at discharge. Daily doses of each insulin regimen showed no differences between two hospitals. The bolus/basal ratio in basal-and-blous insulin treatment didn't differ either.

## Comparison of patient's demographical and clinical characteristics among different discharge pharmacotherapies at LHLH and SMH

Clinical characteristics were then analyzed based on different discharge pharmacotherapies at both hospitals (Tables 3–5). As shows in Table 3, at LHLH, compared with patients on OADs-only treatment, those on insulin treatment had lower BMI, longer diabetes duration and higher admission HbA1c concentration. More patients were complicated with macro- or micro- vascular diseases. Insulin treatment is related to prolonged hospital LOS (8.6±3.7 vs 7.6 ±3.8, days, $p<0.01$). Insurance coverage and BMI were the main differences among OADs regimens at LHLH. More patients with least insurance coverage were given sulfonylurea and patients receiving incretin therapy had highest BMI. BMI differs significantly among different insulin regimens at LHLH, with the lowest BMI in basal-and-bolus insulin treatment, the highest in basal insulin treatment, and premix insulin treatment falling in between.

**Table 3. Patients' clinical characteristics among different discharge pharmacotherapies at LHLH.**

| | OADs-only treatment | | | | | | | Insulin treatment | | | | | | | |
|---|---|---|---|---|---|---|---|---|---|---|---|---|---|---|---|
| | Incretin-based | *SUs-based | Glinide-based | **Nonsecretagogue-based | Total | t/X² | P value | Basal insulin | Premix insulin | Bolus/basal insulin | Total | t/X² | P value | t/X² | P value |
| N (%) | 49 (28) | 72 (41.1) | 5 (2.9) | 49 (28) | 175 (100) | | | 93 (43.1) | 84 (38.9) | 39 (18) | 216 (100) | | | | |
| Age (years) | 56.5 ±15.0 | 56.0 ±11.8 | 65.6 ±10.4 | 60.0±11.2 | 57.5 ±12.7 | 1.78 | 0.15 | 58.0 ±12.4 | 61.4 ±11.1 | 57.7 ±12.2 | 59.6 ±11.9 | 2.23 | 0.11 | 1.37 | 0.17 |
| Male (n, %) | 29 (59.2) | 44 (61.1) | 4 (80.0) | 25 (51.0) | 102 (58.3) | 2.29 | 0.52 | 53 (57.0) | 48 (57.1) | 26 (66.7) | 127 (58.8) | 1.22 | 0.54 | 0.01 | 0.92 |
| Illiterate (n, %) | 5 (10.2) | 9 (12.5) | 0 (0.0) | 13 (26.5) | 27 (15.4) | 7.04 | 0.11 | 13 (14.0) | 10 (11.9) | 4 (10.3) | 27 (12.5) | 0.39 | 0.82 | 0.7 | 0.4 |
| Insurance-covered (n, %) | 35 (71.4) †‡ | 24 (33.3) † | 2 (40.0) | 33 (67.5) †† | 94 (53.7) | 22.3 | <0.01 | 60 (64.5) | 48 (57.1) | 20 (51.3) | 128 (59.3) | 2.25 | 0.33 | 1.21 | 0.27 |
| BMI (kg/m²) | 27.1 ±4.41 † | 23.3 ±2.89 | 21.6 ±0.57 † | 24.4±4.33 | 24.8 ±4.14 | 7.399 | <0.01 | 24.3 ±2.76 †‡ | 22.6 ±3.12 †‡ | 21.3 ±3.11 †‡ | 23.1 ±3.16 | 12.02 | <0.01 | 4 | <0.01 |
| Hospital LOS (days) | 7.6±3.33 | 7.2 ±3.33 | 6.2±1.10 | 8.5±4.81 | 7.6 ±3.79 | 0.772 | 0.511 | 8.7 ±3.99 | 8.9±3.70 | 7.9±3.26 | 8.6 ±3.75 | 0.901 | 0.408 | -2.63 | <0.01 |
| Duration (years) | 6.4±7.32 | 7.7 ±6.11 | 8.8±6.24 | 8.4±7.49 | 7.5 ±6.86 | 1.4 | 0.245 | 8.9 ±6.94 | 11.9 ±7.63† | 9.3±6.47 | 10.5 ±6.82 | 4.392 | 0.014 | -4.19 | <0.01 |
| Newly diagnosed (n, %) | 10 (20.4) | 15 (20.8) | 4 (80%) | 2 (4.1%) | 41 (23.4) | 17.754 | <0.05 | 16 (17.2) | 9 (10.7) | 8 (20.5) | 33 (15.3) | 3.424 | 0.194 | 4.186 | 0.051 |
| Admission HbA1c (%) | 8.80 ±2.51 | 9.10 ±2.73 | 8.06 ±2.99 | 8.60±2.81 | 8.84 ±2.69 | 0.48 | 0.7 | 9.98 ±2.32 | 9.86 ±2.71 | 10.96 ±2.66 | 10.11 ±2.56 | 2.64 | 0.07 | -4.71 | <0.01 |
| Hypertension (n, %) | 29 (59.2) | 35 (48.6) | 3 (60.0) | 31 (63.3) | 94 (56.0) | 2.88 | 0.41 | 48 (51.6) | 47 (56.0) | 14 (35.9) | 109 (50.5) | 4.37 | 0.11 | 1.19 | 0.28 |
| Macro-VSD (n, %) | 30 (61.2) | 46 (63.9) | 1 (20.0) | 33 (67.5) | 110 (62.9) | 4.45 | 0.22 | 72 (77.4) | 69 (82.1) | 29 (74.4) | 170 (78.7) | 1.25 | 0.54 | 11.94 | <0.01 |
| Micro-VSD (n, %) | 3 (6.1) | 0 (0) | 1 (20.0) | 1 (2.0) | 5 (2.9) | 6.48 | 0.09 | 25 (27.0) | 28 (33.3) | 12 (33.8) | 24 (30.1) | 0.88 | 0.64 | 13.58 | <0.01 |

*patients with DPP-4i + sulfonylurea were included in 'SUs-based'

** patients with diet-only were included in 'nonsecreatagogue-based'

†significantly different compared with other groups

†‡significantly different between groups; macro-VSD, macro-vascular diseases (peripheral artery diseases: carotid artery atherosclerosis and/or lower limb artery atherosclerosis, coronary heart disease, cerebral vascular diseases); micro-VSD, micro-vascular disease (diabetic kidney diseases and/or diabetic retinopathy).

Table 4 shows the clinical characteristics of patients on different discharge pharmacotherapies at SMH. Admission HbA1c concentration seemed to be the sole differentiating factor among discharge pharmacotherapies at SMH. There were less newly diagnosed diabetes among patients on insulin treatment. Hospital LOS was longer for insulin treatment. Clinical characteristics didn't significantly differ among OADs-only treatment at SMH. Basal-and-bolus insulin treatment had the highest admission HbA1c concentration.

Generalized linear models with mixed effects including 'hospital' were used to analyze the differences between treatments (OADs-only treatment and insulin treatment) with 'hospital' turning out to be a significant impact on treatment (OR: 0.352, 95%CI [0.213, 0.570]). GLM was then extended to generalized linear mixed models (GLMM) with 'hospital' incorporated as the random effect and the other variables as fixed effects to study differences between treatments. Overall higher admission HbA1c concentration, longer duration of diabetes and vascular complications were predictive of insulin treatment (results showed in Table 5).

**Table 4. Patients' clinical characteristics among different discharge pharmacotherapies at SMH.**

| | OADs-only treatment | | | | | | | Insulin treatment | | | | | | | |
| --- | --- | --- | --- | --- | --- | --- | --- | --- | --- | --- | --- | --- | --- | --- | --- |
| | Incretin-based | SUs-based* | Glinide-based | Nonsecretagogue-based | Total | t/$X^2$ | P value | Basal insulin | Premix insulin | Basal-bolus insulin | Total | t/$X^2$ | P value | t/$X^2$ | P value |
| N (%) | 2 (4.7) | 25 (58.1) | 7 (16.3) | 9 (20.9) | 43 (100) | | | 38 (31.4%) | 10 (8.3%) | 73 (60.3) | 121 (100) | | | | |
| Age (years) | 57.0±5.66 | 62.8 ±9.40 | 61.0 ±11.26 | 62.6±12.52 | 62.0 ±9.92 | 0.236 | 0.871 | 59.5 ±11.01 | 68.3 ±8.53 | 60.7 ±10.82 | 61.0 ±10.88 | 2.737 | 0.069 | 0.554 | 0.581 |
| Male (n, %) | 0 | 12 (48.0) | 2 (28.6) | 4 (44.4) | 18 (41.9) | 2.36 | 0.616 | 12 (31.6) | 5 (50.0) | 33 (45.2) | 50 (41.3) | 2.296 | 0.306 | 0.131 | 0.717 |
| Illiterate (n, %) | 1 (50) | 16 (64.0) | 2 (28.6) | 5 (55.6) | 24 (55.8) | 1.471 | 0.777 | 18 (47.4) | 7 (70.0) | 39 (53.4) | 64 (52.9) | 1.595 | 0.458 | 4.238 | 0.054 |
| Insurance-covered (n, %) | 1 (50) | 23 (92) | 5 (71.4) | 9 (100.0) | 38 (88.4) | 5.825 | 0.062 | 35 (92.1) | 9 (90.0) | 61 (83.6) | 105 (86.8) | 1.496 | 0.454 | 8.735 | 0.003 |
| Body weight (kg) | 56.0 | 67.30 ±12.25 | 63.5 ±12.72 | 70.17±17.08 | 67.08 ±13.20 | 0.435 | 0.73 | 64.11 ±10.58 | 65.82 ±12.68 | 62.55 ±8.93 | 63.32 ±9.77 | 0.578 | 0.563 | 1.534 | 0.128 |
| Hospital LOS (days) | 9.5±3.54 | 7.5 ±3.64 | 7.6±2.76 | 6.6±2.46 | 7.4 ±3.24 | 0.483 | 0.696 | 8.2 ±2.40 | 8.4±3.03 | 8.7±2.79 | 8.7 ±2.79 | 1.036 | 0.358 | -2.236 | 0.027 |
| Duration (years) | 9.3±12.37 | 10.5 ±10.22 | 7.8±6.54 | 8.3±7.76 | 9.55 ±9.07 | 0.227 | 0.877 | 9.8 ±7.04 | 15.7 ±7.87 † | 7.7±7.61 | 9.0 ±7.72 | 5.315 | 0.006 | 0.37 | 0.712 |
| Newly diagnosed (n, %) | 1 (50) | 16 (64.0) | 5 (71.4) | 4 (44.4) | 19 (44.2) | 3 | 0.405 | 16 (42.1) | 0 | 19 (26.0) | 35 (28.9) | 7.69 | 0.02 | 103.78 | <0.05 |
| Admission HbA1c (%) | 8.35±0.78 | 9.38 ±1.60 | 10.11 ±2.41 | 9.20±0.85 | 9.42 ±1.61 | 0.774 | 0.516 | 10.03 ±1.92 | 10.62 ±2.19 | 11.23 ±2.03 † | 10.80 ±2.07 | 4.502 | 0.01 | -4.213 | <0.01 |
| Hypertension (n, %) | 1 (50.0) | 13 (52.0) | 3 (42.9) | 6 (66.7) | 23 (53.5) | 0.98 | 0.87 | 26 (68.4) | 7 (70) | 46 (63.0) | 79 (65.3) | 0.42 | 0.84 | 1.99 | 0.16 |
| Macro-VSD (n, %) | 1 (50.0) | 6 (24.0) | 1 (14.3) | 1 (11.1) | 10 (22.2) | 2.05 | 0.63 | 18 (47.4) | 4 (40.0) | 29 (26.0) | 51 (38.9) | 1.76 | 0.43 | 4.13 | 0.047 |
| Micro-VSD (n, %) | 1 (50.0) | 0 (0) | 0 (0) | 0 (0) | 2 (4.4) | 21 | 0.047 | 3 (7.9) | 3 (30.0) | 6 (8.2) | 12 (9.9) | 4.16 | 0.13 | 1.27 | 0.23 |

*patients with DPP-4i + sulfonylurea were included in 'SUs-based'

†significantly different compared with other groups †‡signifiScantly different between groups

macro-VSD, macro-vascular diseases (peripheral artery diseases: carotid artery atherosclerosis and/or lower limb artery atherosclerosis; coronary heart disease, cerebral vascular diseases); micro-VSD, micro-vascular disease (diabetic kidney diseases, diabetic retinopathy).

**Table 5. Predictors of insulin treatment as discharge pharmacotherapy studied in GLMM.**

| | Value | Std.Error | t-value | p-value | OR | 95% CI |
| --- | --- | --- | --- | --- | --- | --- |
| Intercept | -2.457 | 0.744 | -3.302 | <0.001 | 0.151 | [0.02, 0.42] |
| HbA1c | 0.306 | 0.047 | 6.473 | <0.001 | 1.354 | [1.24, 1.49] |
| duration | 0.0444 | 0.015 | 2.979 | 0.003 | 1.045 | [1.02, 1.08] |
| Micro-VSD | | | | | | |
| yes | 2.583 | 0.457 | 5.646 | <0.001 | 13.24 | [5.82, 35.95] |
| Macro-VSD | | | | | | |
| yes | 0.493 | 0.231 | 2.136 | 0.033 | 1.64 | [1.04, 2.59] |

[With an Akaike information criterion, AIC value: 606.7 ]

## Discussion

Pharmacotherapy for T2DM not complicated by factors like comorbidities, organ dysfunctions, pregnancies or glucose-interfering medications in two hospitals were studied. The exclusion criteria in this study were based on two considerations: firstly we seek to exclude situations that limit pharmaceutical choices, either contraindicating or predefining certain anti-diabetics [2, 9]. As a fact, patients with aforementioned conditions are most possibly referred to tertiary hospitals. Secondly it's more expedient if we mean to inspect how the standardized guideline for T2DM management is carried out at hospitals of different ranks. So the study zooms in on uncomplicated T2DM requiring intensification of their diabetes therapy. Despite the fact that the study was carried out among inpatients, some findings could be cautiously extrapolated to the outpatients as well.

Not unexpected with our understanding of disease status in rural and urban populations [1, 19, 21], patients' characteristics from two hospitals showed significant differences. Patients from SMH were older and less educated. The discordance in age may have caused the female preponderance in patients from SMH [1]. Though it's not a population-based study, the higher proportion of newly diagnosed and untreated diabetes, higher concentration of admission HbA1c still suggest delayed detection of and treatment for diabetes and poorer control of blood glucose in the rural area [21]. Despite the poorly controlled hyperglycemia, less than half of patients in our study had initiated insulin treatment compared with results from the former study which was carried out in T2DM outpatients [22]. In another word, delayed insulin initiation at least partly accounts for the poor control of glycaemia even in patients from LHLH in our study and the situation in rural areas is even worse as showed in patients from SMH. Insulin initiation inertia is still a common problem in inadequately controlled T2DM patients [23–25].

We noticed that a pattern of difference exists in the diagnosing of comorbidities at two hospitals. While the prevalence of hypertension, the diagnosing of which requires of no sophisticated equipment or techniques, is much higher at SMH, conditions diagnosed by routine screening (like atherosclerotic artery disease), symptoms/high risks and targeted screening (like coronary heart disease), or by medical history (like cerebral vascular diseases) at LHLH showed a stepwise narrower discrepancy in prevalence compared with those at SMH. It implies that chronic vascular complications usually get diagnosed only at advanced stages in primary institutions.

With the baseline clinical characteristics significantly different between two hospitals, we didn't directly compare their discharge pharmacotherapy arrangements. But some distinctions are still apparent. Generally speaking, discharge pharmacotherapy at LHLH manifests principal considerations in diabetes management, like considerations of weight control and cost besides blood glucose [2, 9] (Table 3). Whereas management at SMH is mostly glucose-oriented (Table 4). Limited availability and less familiarity of newer classes of anti-diabetics at SMH partly explain the discrepancy. Delayed evaluation/diagnosis of vascular complications also accounts for the fact since these medications are exactly the options if beyond-glucose benefits (e.g. weight control, improvement of cardiovascular outcomes) are expected of [2, 9, 12].

Quite contrary to the comparison of admission diabetes therapy between LHLH and SMH, insulin treatment dominated the discharge pharmacotherapy at SMH while OADs-only treatment and insulin contributed equally at LHLH. No significant differences were noticed in the utilization of traditional OADs like metformin and sulfonylurea. Nearly 70% of all patients were receiving metformin, which is consistent with its recognized first-line role in pharmacological management of T2DM [2, 9, 12] and about one-quarter of patients were receiving

sulfonylurea at both hospitals. AGI seemed to be more commonly used at LHLH which could partly be explained by the different patterns of insulin treatment in two hospitals since AGI is more often combined with other anti-diabetics including basal insulin treatment. TZD appeared to be underutilized at both hospitals (yet considering the daily doses of insulin in this study, our patients did appear to not so apparently insulin resistant). The most distinctive difference in OADs-only treatment between hospitals exists in the utilization of newly developed medications with innovative mechanisms. It is still rare that DPP4i/GLP-1RA/SGLT2i be prescribed at SMH. So it takes time and requires continuous multi-sectoral efforts (e.g. clinical practice of evidence-based medicine, drug pricing and reimbursement, and so on).

The most worrying result is the preponderance basal-and-bolus (BBI) insulin treatment assumed at SMH's discharge pharmacology. While nearly three-quarters of the patients from SMH were given insulin treatment on discharge, most of them remained on BBI. The insulin regimens at LHLH were quite oppositely distributed, with BBI assuming only 18%, basal insulin 43.1% and premix insulin 38.9%. The insulin regimen distribution in LHLH's discharge pharmacotherapy is more consistent with the recognized status of insulin treatment at discharge for T2DM as the addition of basal insulin being the most frequent and effective modification of treatment [16] and premix insulin formula an non-inferior choice to BBI [26]. We tried to unravel the disparities in the following discussion.

At first, the different POC (point of care) blood glucose (BG) testing patterns at two hospitals. At LHLH, fasting, pre-meal and bedtime POCT of BG are the routine for inpatient BG monitoring, whereas at SMH fasting BG, 2h-postprandial BG after each main meal and bedtime BG monitoring are the routine. Inpatient management of hyperglycemia is based on intensive BG monitoring [14, 27]. Though Chinese guideline doesn't specify BG monitoring and dose-adjustment in its insulin treatment path [2], multiple consensus on insulin treatment generally follow certain procedures in adjusting insulin doses [14, 27–29]. Fasting BG in the morning is usually the baseline BG key to the adjustment of basal insulin dose, followed by non-fasting BG (usually represented by pre-meal BG) to get an overall evaluation of suitability of total daily dose (TDD) of insulin. Basal insulin is crucial to overall BG homeostasis and prandial insulin gets reasonable titration only on the basis of a reasonable basal insulin dose. Most of the inpatient hyperglycemia managements recommend monitoring of pre-meal BG over postprandial BG as reference for dose adjustment [27, 29]. Routine monitoring of postprandial BG at SMH, especially before the overall hyperglycemic state has got under control, is highly possible to influence the inpatient insulin treatment and discharge planning since greater variability of postprandial hyperglycemia could be taken as an indicator of BBI treatment.

Secondly, significant postprandial hyperglycemia in our population [1]. Compared with the white ethnics, Chinese diabetics are more β cell deficiency-oriented [30, 31]. Deficient first-phrase insulin secretion in early β cell dysfunction causes great postprandial hyperglycemia excursion [30] and Chinese carbohydrate-rich diet fuels the flame [32, 33]. Epidemiology studies repeatedly verified this characteristic of the population's blood glucose pattern [1, 21]. TDD of insulin, and bolus/basal ratio in our findings also indicate that more therapeutic emphasis be put on postprandial hyperglycemia (Table 2). The CLASSIFY study shows mid mixture insulin formula (Humalog® Mix 50, LM50) better suits patients with higher PPG and carbohydrate-rich diet than low mixture insulin formula (Humalog® Mix 25, LM 25) in our population [34]. Yet mid mixture of insulin formulations are usually unavailable in secondary institutions like SMH. This may be related to the fact that more BBI were chosen over premix insulin at SMH.

Thirdly, the more common utilization of incretin-based therapy, especially GLP-1RA, at LHLH. The combination of GLP-1RA and basal insulin has verified efficacy in lowering blood glucose and has extra advantage in weight control and lower risk of hypoglycemia. As a result,

basal insulin plus GLP-1RA is a guideline-supported combination that may replace BBI in many patients with satisfying therapeutic response [9, 11, 12]. Yet GLP-1RA is of limited availability at SMH and primary practitioners usually are less familial with new classes of OADs. Treatment inertia also tends to happen in utilizations of unfamiliar drugs [25]. Cost may be another reason. Even with insurance coverage, new cooperative medical scheme (NCMS) is the most prevalent medical insurance type for the rural population. NCMS is a new and basic type of social medical insurance with least qualification for applying and is inferior in its coverage and rate of reimbursement compared with other types of medical insurance (namely, the urban employee basic medical insurance and urban resident basic medical insurance). So it still appears quite a financial burden to the rural population if they accept long-term treatment with GLP-1RA. Urban populations, in contrast, usually are more willing to invest in personal health insurance plans other than the basic employee/residence medical insurance for more medical security. Our medical records system usually doesn't record these personal insurance plans since patients get reimbursement after their discharge. So it's not reflected in our data on insurance-coverage, which may under-estimate the disadvantage in insurance-coverage for the rural population as well.

The last but not the least, lack of discharge transition plans. BBI is generally recommended for inpatient hyperglycemia [14–16] but it's usually not a practical choice for T2DM outpatients. The complexity of BBI is not just the intensity of multiple injections but the need of intensified monitoring of blood glucose for dose adjustment, the possibility of mistaken injection with the wrong insulin (since basal insulin and prandial insulin need to be administrated separately via two alike injection devices), necessary matching of meal and prandial insulin (carbohydrate counting), unintentional omission of insulin injections or negligence of meals after injections ('Injection- without-meal' or 'meal-without-injection') [28, 35]. The situation could be worse in the old population with decreased capability of disease management (declined cognitive ability [36], decreased manual dexterity [37], poor sight [2] and so on) [38]. Medication compliance drops post-discharge almost inevitably, especially in insulin treatment [39]. Even with a standard discharge algorithm and close post-discharge follow-up, BBI therapy has the highest hypoglycemia risk [17]. It's prudent to be concerned that, with the more advanced age, lower level of literacy and poorer compliance with disease management, patients from SMH may have multiple barriers to safely implement BBI treatment post-discharge [38]. There were studies on feasible algorithms for switch from BBI treatment to basal insulin or premix insulin or OAD-only treatment before discharge [16, 17]. As a fact, a similar algorithm is applied to arrange patients' discharge pharmacotherapy at LHLH, which gives a totally different pattern in distribution of discharge medication. Pitifully the Chines guideline puts little emphasis on this part. There are little recommendations or any process for discharge transition [2], and even less about tailored discharge pharmacotherapy for T2DM [14]. Developing practical algorithms for therapeutic transition from hospitalization to home management is of paramount importance [40]. We hold that a cautious inference could be made from our analysis on the influences of discharge pharmacology (Tables 3 and 5) that a practical algorithm based on HbA1c to direct discharge pharmacology (especially in insulin initiation and adjustment) in inadequately controlled T2DM is feasible and incorporation of disease characteristics like duration and vascular conditions might provide extra optimization.

## Limitations

There are limitations in the study. First of all, patients from two hospitals are not comparable in their demographical and clinical features and we didn't stratify patients to make comparable analyses. Secondly, Different therapeutic management of diabetes during the hospital stay

could induce variable therapeutic responses and influence discharge pharmacology [41]. As a truth, continuous subcutaneous insulin infusion via insulin pump is a common choice for hyperglycemia management at LHLH but is rarely used at SMH. But the analysis doesn't include diabetes treatment during the hospital stay, hence it's impossible to analyze its effects on the discharge pharmacotherapy at two hospitals. Then we didn't employ strict diagnosing and classification criteria for chronic diabetic complications in the study considering the limited data at SMH. We also excluded patients with advanced stages of DKD and DR, who have developed overt kidney dysfunction or may require ophthalmologic operation for vitreous hemorrhage, for example, so prevalence of DKD and DR in this study must be lower than other studies [22, 42, 43]. Hence the study is not for analyzing prevalence of diabetic complications. Restricted by the retrospective nature of data collection in this analysis, we couldn't absolutely eliminate the possibility of omission or even some erroneous information, either. In the end, our analysis are performed on a particular group of inadequately controlled T2DM in one health alliance, hence is limited in its scope and generalizability. Discharge planning is with no doubt a commonly overlooked part of nowadays diabetes management and future population-based studies are imperative to improve generalizability to other clinical populations as well as geographic regions.

## Conclusions

In conclusion, great disparities exist in two hospitals' discharge pharmacotherapy for T2DM. Lack of practical algorithm for discharge transition of pharmacological arrangements is probably the most critical deficiency currently exists in hospitals of lower ranks. No efforts are to be spared, either, in implementing standardized care to the insulin initiation and intensification, especially in the rural population.

## Acknowledgments

Our deepest gratitude goes to the supervising office of Medical Alliance since their sustaining support are key for the whole project. Thank all colleagues at two hospitals for their enthusiastic contributions to the construction and improvement of medical alliance and trusteeship between two hospitals.

## Author Contributions

**Conceptualization:** Xiaofang Yu, Long Zhang, Jiao Yang.

**Data curation:** Long Zhang, Rongbin Yu, Jiao Yang, Saifei Zhang.

**Formal analysis:** Xiaofang Yu, Rongbin Yu.

**Investigation:** Xiaofang Yu, Long Zhang, Jiao Yang, Saifei Zhang.

**Methodology:** Xiaofang Yu, Long Zhang.

**Project administration:** Xiaofang Yu, Jiao Yang.

**Software:** Rongbin Yu.

**Supervision:** Xiaofang Yu.

**Validation:** Xiaofang Yu.

**Visualization:** Xiaofang Yu.

**Writing – original draft:** Xiaofang Yu.

**Writing – review & editing:** Xiaofang Yu.

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
