## [Decision Letter · Decision Letter 0]

16 Dec 2019

PONE-D-19-28681

Discharge Pharmacotherapy for Type 2 Diabetic Inpatients in two hospitals of different tiers in Zhejiang Province, China

PLOS ONE

Dear Ms Yu,

Thank you for submitting your manuscript to PLOS ONE. After careful consideration, we feel that it has merit but does not fully meet PLOS ONE’s publication criteria as it currently stands. Therefore, we invite you to submit a revised version of the manuscript that addresses the points raised during the review process.

We would appreciate receiving your revised manuscript by Jan 27 2020 11:59PM. To enhance the reproducibility of your results, we recommend that if <gwmw class="ginger-module-highlighter-mistake-type-3" id="gwmw-15762289649851686342520">applicable you</gwmw> deposit your laboratory protocols in protocols.io, where a protocol can be assigned its own identifier (DOI) such that it can be cited independently in the future. For <gwmw class="ginger-module-highlighter-mistake-type-3" id="gwmw-15762289653791719801840">instructions see</gwmw>: http://journals.plos.org/plosone/s/submission-guidelines#loc-laboratory-protocols

A rebuttal letter that responds to each point raised by the academic editor and reviewer(s). This letter should be uploaded as <gwmw class="ginger-module-highlighter-mistake-type-3" id="gwmw-15762289667721487976821">separate file</gwmw> and labeled 'Response to Reviewers'.A marked-up copy of your manuscript that highlights changes made to the original version. This file should be uploaded as <gwmw class="ginger-module-highlighter-mistake-type-3" id="gwmw-15762289681214050094729">separate file</gwmw> and labeled 'Revised Manuscript with Track Changes'.An unmarked version of your revised paper without <gwmw class="ginger-module-highlighter-mistake-type-3" id="gwmw-15762289687166675086841">tracked</gwmw> changes. This file should be uploaded as <gwmw class="ginger-module-highlighter-mistake-type-3" id="gwmw-15762289693076713563702">separate file</gwmw> and labeled 'Manuscript'.

Please <gwmw class="ginger-module-highlighter-mistake-type-3" id="gwmw-15762289702592494055739">note while</gwmw> forming your response, if your article is accepted, you may have the opportunity to make the peer review history publicly available. The record will include editor decision letters (with reviews) and your responses to reviewer comments. If eligible, we will contact you to opt in or out.

We look forward to receiving your revised manuscript.

Kind regards,

Manal S. <gwmw class="ginger-module-highlighter-mistake-type-1" id="gwmw-15762289730936842576340">Fawzy</gwmw>, Ph.D., M.D.

Academic Editor

PLOS ONE

Journal Requirements:

2. In the ethics statement in the manuscript and in the online submission form, please provide additional information about the patient records used in your retrospective study, including: a) the date range (month and year) during which patients' medical records were accessed and b) the source of the medical records analyzed in this work (e.g. hospital, institution or medical center name).

3. Please amend your authorship list in your manuscript file to include Long Zhang, Rongbin Yu, Jiao Yang and Saifei Zhang.

"the funders had no role in study design, data collection and analysis, decision to publish, or the preparation of the manuscript."

Please provide an amended Funding Statement that declares *all* the funding or sources of support received during this specific study (whether external or internal to your organization) as detailed online in our guide for authors at http://journals.plos.org/plosone/s/submit-nowPlease state what role the funders took in the study.  If any authors received a salary from any of your funders, please state which authors and which funder. If the funders had no role, please state: "The funders had no role in study design, data collection and analysis, decision to publish, or preparation of the manuscript."

Reviewers' comments:

Reviewer's Responses to Questions

**Comments to the Author**

1. Is the manuscript technically sound, and do the data support the conclusions?

Reviewer #1: Partly

Reviewer #2: Yes

2. Has the statistical analysis been performed appropriately and rigorously? 

Reviewer #1: Yes

Reviewer #2: N/A

3. Have the authors made all data underlying the findings in their manuscript fully available?

The PLOS Data policy requires authors to make all data underlying the findings described in their manuscript fully available without restriction, with rare exception (please refer to the Data Availability Statement in the manuscript PDF file). The data should be provided as part of the manuscript or its supporting information, or deposited <gwmw class="ginger-module-highlighter-mistake-type-3" id="gwmw-15762289847513682069916">to</gwmw> a public repository. For example, in addition to summary statistics, the data points behind means, medians and variance measures should be available. If there are restrictions on publicly sharing data—e<gwmw class="ginger-module-highlighter-mistake-type-3" id="gwmw-15762289862450161392451">.</gwmw>g. <gwmw class="ginger-module-highlighter-mistake-type-1" id="gwmw-15762289867846312597415">participant</gwmw> privacy or use of data from a third party—those must be specified.

Reviewer #1: Yes

Reviewer #2: Yes

4. Is the manuscript presented in an intelligible fashion and written in standard English?

PLOS ONE does not copyedit accepted manuscripts, so the language in <gwmw class="ginger-module-highlighter-mistake-type-3" id="gwmw-15762289883676650415365">submitted</gwmw> articles must be clear, correct, and unambiguous. Any typographical or grammatical errors should be corrected at revision, so please note any specific errors here.

Reviewer #1: No

Reviewer #2: Yes

5. Review Comments to the Author

Please use the space provided to explain your answers to the questions above. You may also include additional comments for the author, including concerns about dual publication, research ethics, or publication ethics. <gwmw class="ginger-module-highlighter-mistake-type-3" id="gwmw-15762289918559247982364">(</gwmw>Please upload your review as an attachment if it exceeds 20,000 characters)

Reviewer #1: This article assessed the service use among Type II diabetes in two different levels of hospitals in Ningbo, Eastern China. While understanding the service use is important, quite many articles have already published in this field. For the selected hospitals in the article, it is also difficult to argue the disparity of service used among patients, since the tertiary may treat more patients with severe conditions or complications. In addition, the impact of the hospital alliance is uncertain.

The research design is <gwmw class="ginger-module-highlighter-mistake-type-3" id="gwmw-15762289954533705014659">valid but</gwmw> the English writing <gwmw class="ginger-module-highlighter-mistake-type-3" id="gwmw-15762289954536380041935">need</gwmw> improvement

Abstract

It is not clear to me, are authors want to detect the deficiencies in discharge management by comparison between the two hospitals?

Is this analysis performed on patients’ medical records? How to identify eligible patients? What time is this analysis conducted?

Introduction

Is there any published clinical guideline for diabetes in China? And its implementation?

A proper literature review <gwmw class="ginger-module-highlighter-mistake-type-3" id="gwmw-15762289999159168624456">need</gwmw> to be conducted and describe the additional value of this article to published work.

Methods:

Patients were included from the two hospitals from different year. Will this influence results?

Any calculation to get the sample size?

If <gwmw class="ginger-module-highlighter-mistake-type-3" id="gwmw-15762290021864362813420">hospital</gwmw> was input into the regression model as <gwmw class="ginger-module-highlighter-mistake-type-3" id="gwmw-15762290021868852976223">dummy variable</gwmw>, what will happen? Will “hospital” have significance in treatment when control other variables?

Reviewer #2: The authors conducted a retrospective cross-sectional study of the discharge <gwmw class="ginger-module-highlighter-mistake-type-1" id="gwmw-15762290042337842522572">pharmacotherapy</gwmw> for type 2 diabetes <gwmw class="ginger-module-highlighter-mistake-type-1" id="gwmw-15762290042331134042904">inpatients</gwmw> in their two hospitals of different tiers. Data is relevant for publication, since there doesn’t exist similar information, especially in a new type of medical referral system named “Regional Longitudinal Health Alliance” in China. The knowledge of the <gwmw class="ginger-module-highlighter-mistake-type-1" id="gwmw-15762290061832918627919">pharmacotherapy</gwmw> in this particular population and setting could be used by Chinese experts in developing, evaluating implementing interventions for optimizing diabetes managements.

In general, I appreciated this manuscript. Although it is conducted in a scientifically meaningful way, there are still some limitations which should be clarified.

1. Line90-93: Please provide more information on the “Regional Longitudinal Health Alliance”. For example, how the hospitals of different tiers in the Health Alliance interact with each other? Is this just a referral system? How does this Health Alliance improve the management of T2DM?

2. Line97-99: I recommend to delete this sentence. Although patients can skip the referral system and choose any hospitals now, I think it will take time to implement as new rules.

3. Line91: Please provide more information on tertiary hospital, secondary hospital and community <gwmw class="ginger-module-highlighter-mistake-type-1" id="gwmw-15762290110225647029859">healthcare</gwmw> centers.

4. Line125: Please clarify the definition of “clinically relevant hepatic disease”.

5. Line128: Please readjust the serial number of exclusion criteria, which is a small but obvious mistake.

6. Line 128-134: I recommend to move these sentences to the discussion, as they are not part of the methods.

7. Line 140: Please clarify the definition of “cerebral vascular disease”. Ischemic stroke or hemorrhagic stroke?

8. Line 138: In table 1, you mentioned the variable “newly diagnosed diabetes”, but it was not mentioned in the methods, please add it and clarify the definition of “newly diagnosed diabetes”.

9. Line155-156: Has informed consent <gwmw class="ginger-module-highlighter-mistake-type-3" id="gwmw-15762290169034161095588">been obtained</gwmw> by participants? Probably not, as there was no access to individual data by the first author/corresponding author. Please indicate so in methods and in Line155.

10. Line140-141: Please provide more information of the diagnostic process of the micro- and macro-vascular complications.

11. Line 153-154: Please provide more information of the regression analysis. For example, which independent variables were selected in the regression analysis?

12. Line 164-165: Please recheck the data listed in this sentence and in table 1.

13. Line 173 table 1: Please recheck the data in table 1. There is <gwmw class="ginger-module-highlighter-mistake-type-3" id="gwmw-15762290230800552441407">88</gwmw><gwmw class="ginger-module-highlighter-mistake-type-3" id="gwmw-15762290230807918417339">(</gwmw>53.1) in the row of Illiterate and the column of SMH, 74<gwmw class="ginger-module-highlighter-mistake-type-3" id="gwmw-15762290230804369899832">(</gwmw>91.4) in the row of Insurance-covered and the column of SMH. You should double-check these data whether they are right or not.

14. Line 173 table 1: In the row of BMI in table 1, please maintain 2 digits at most after <gwmw class="ginger-module-highlighter-mistake-type-3" id="gwmw-15762290249752826860713">decimal point</gwmw>. Same as in table 3 and table 4.

15. Line 173 table 1: Patients in both hospitals have a long duration (>9 years) and poor glycemic control (HbA1c>9.5%), but the incidence of diabetic micro-vascular complications is relevant low, especially in SMH. Please discuss.

16. Line 177: A total of 164 T2DM patients <gwmw class="ginger-module-highlighter-mistake-type-3" id="gwmw-15762290280485349648624">were included</gwmw> in this study. In table 2, the authors reported 43 patients were treated with OAD, 121 patients were treated with insulin. The total number was 164. In line 177, the authors said that “only 1 patient was treated with diet only in SMH”. Please double-check the data “only 1 patient (0.6%)” whether they are right or not.

17. Line 179-181: In table 2, we did not find a significant difference in OADs between two hospitals (P=0.34). So please rephrase this sentence.

18. Line 251: You wrote about less patients in our study had initiated insulin treatment-however, it does not <gwmw class="ginger-module-highlighter-mistake-type-2" id="gwmw-15762290333426611192739">become clear less</gwmw> than what/where? Do you refer to the <gwmw class="ginger-module-highlighter-mistake-type-3" id="gwmw-15762290339895989877758">previously</gwmw> figures in the other study? If so, please give numbers for this statement as well.

19. Line288: Similar to my comment 18. Please add more information.

20. Line 324-330: Although patients in rural area <gwmw class="ginger-module-highlighter-mistake-type-3" id="gwmw-15762290370874008942860">has</gwmw> the NCMS, they can’t afford the financial burden because such new type of medications <gwmw class="ginger-module-highlighter-mistake-type-3" id="gwmw-15762290370870417318990">are not covered</gwmw> by NCMS. But in LHLH, only 57.5% of all subjects were covered by insurance, why these people were prescribed with more GLP-1A? Please discuss.

21. Line333-340: Please add some references.

22. I can’t find table 5 was discussed or provide any useful information in the manuscript, if this is true, I recommend to delete this table.

23. Line353: In terms of <gwmw class="ginger-module-highlighter-mistake-type-1" id="gwmw-15762290409325191724073">generalizability</gwmw>, this study only included T2DM inpatients in two specific hospitals, the findings could be valid in similar populations and settings. Inpatients have higher blood glucose or sever conditions. A population-based study should be conducted, which might provide a different picture of the issue. Please add this in limitations section.

While in general, I like your manuscript. It delivers valuable information, there are still some nooks and crannies that need improvement. I think some obvious mistakes, especially data errors, should be avoided.

Thank you and good luck!

6. PLOS authors have the option to publish the peer review history of their article (what does this mean?). If published, this will include your full peer review and any attached files.

If you choose “no”, your identity will remain <gwmw class="ginger-module-highlighter-mistake-type-3" id="gwmw-15762290468974574144994">anonymous but</gwmw> your review may still be made public.

Reviewer #1: No

Reviewer #2: No

<gdiv></gdiv>

---

## [Author Response · Author response to Decision Letter 0]

6 Feb 2020

Dear Reviewers

 Thank you very much for your meticulous work. We've made our revision based on your enlightening suggestions and addressed your generous comments point by point. We believe it has greatly impoved our work.

---

## [Decision Letter · Decision Letter 1]

24 Feb 2020

Discharge Pharmacotherapy for Type 2 Diabetic Inpatients at two hospitals of different tiers in Zhejiang Province, China

PONE-D-19-28681R1

Dear Dr. Yu,

We are pleased to inform you that your manuscript has been judged scientifically suitable for publication and will be formally accepted for publication once it complies with all outstanding technical requirements.

Shortly after the formal acceptance letter is sent, an invoice for payment will follow. To ensure an efficient production and billing process, please log <gwmw class="ginger-module-highlighter-mistake-type-1" id="gwmw-15823316201520394534216">into</gwmw> Editorial Manager at https://www.editorialmanager.com/pone/, click the "Update My Information" link at the top of the page, and update your user information. If you have any billing related questions, please contact our Author Billing department directly at authorbilling@plos.org.

With kind regards,

Manal S. <gwmw class="ginger-module-highlighter-mistake-type-1" id="gwmw-15823316246459315628498">Fawzy</gwmw>, Ph.D., M.D.

Academic Editor

PLOS ONE

Additional Editor Comments (optional):

The authors have adequately addressed all the concerns raised by the reviewers. Thank you

Reviewers' comments:

Reviewer's Responses to Questions

**Comments to the Author**

1. If the authors have adequately addressed your comments raised in a previous round of review and you feel that this manuscript is now acceptable for publication, you may indicate that here to bypass the “Comments to the Author” section, enter your conflict of interest statement in the “Confidential to Editor” section, and submit your "Accept" recommendation.

Reviewer #1: All comments have been addressed

Reviewer #2: All comments have been addressed

2. Is the manuscript technically sound, and do the data support the conclusions?

Reviewer #1: Yes

Reviewer #2: Yes

3. Has the statistical analysis been performed appropriately and rigorously? 

Reviewer #1: Yes

Reviewer #2: Yes

4. Have the authors made all data underlying the findings in their manuscript fully available?

The PLOS Data policy requires authors to make all data underlying the findings described in their manuscript fully available without restriction, with rare exception (please refer to the Data Availability Statement in the manuscript PDF file). The data should be provided as part of the manuscript or its supporting information, or deposited <gwmw class="ginger-module-highlighter-mistake-type-3" id="gwmw-15823316361672495389353">to</gwmw> a public repository. For example, in addition to summary statistics, the data points behind means, medians and variance measures should be available. If there are restrictions on publicly sharing data—e<gwmw class="ginger-module-highlighter-mistake-type-3" id="gwmw-15823316375412301881217">.</gwmw>g. <gwmw class="ginger-module-highlighter-mistake-type-1" id="gwmw-15823316380608289261817">participant</gwmw> privacy or use of data from a third party—those must be specified.

Reviewer #1: Yes

Reviewer #2: Yes

5. Is the manuscript presented in an intelligible fashion and written in standard English?

PLOS ONE does not copyedit accepted manuscripts, so the language in <gwmw class="ginger-module-highlighter-mistake-type-3" id="gwmw-15823316396086712521817">submitted</gwmw> articles must be clear, correct, and unambiguous. Any typographical or grammatical errors should be corrected at revision, so please note any specific errors here.

Reviewer #1: Yes

Reviewer #2: Yes

6. Review Comments to the Author

Please use the space provided to explain your answers to the questions above. You may also include additional comments for the author, including concerns about dual publication, research ethics, or publication ethics. <gwmw class="ginger-module-highlighter-mistake-type-3" id="gwmw-15823316426029058052575">(</gwmw>Please upload your review as an attachment if it exceeds 20,000 characters)

Reviewer #1: Thanks for authors' efforts in revising the manuscript. My previous comments have been addressed properly and I have no further suggestions.

Reviewer #2: Thank you very much for your extensive update of the manuscript. You have addressed all of my comments in a very sufficient manner.

7. PLOS authors have the option to publish the peer review history of their article (what does this mean?). If published, this will include your full peer review and any attached files.

If you choose “no”, your identity will remain <gwmw class="ginger-module-highlighter-mistake-type-3" id="gwmw-15823316468013309063857">anonymous but</gwmw> your review may still be made public.

Reviewer #1: Yes: Wenhui Mao

Reviewer #2: No

<gdiv></gdiv>

---

## [Editor Report · Acceptance letter]

18 Mar 2020

PONE-D-19-28681R1 

Discharge Pharmacotherapy for Type 2 Diabetic Inpatients at two hospitals of different tiers in Zhejiang Province, China 

Dear Dr. Yu:

I am pleased to inform you that your manuscript has been deemed suitable for publication in PLOS ONE. Congratulations! Your manuscript is now with our production department. 

With kind regards,

on behalf of

Professor Manal S. Fawzy 

Academic Editor

PLOS ONE